# HOW TO GUESS A GRADIENT

## ABSTRACT

How much can you say about the gradient of a neural network without *computing a loss* or *knowing the label*? This may sound like a strange question: surely the answer is "very little." However, in this paper, we show that gradients are more structured than previously thought. Gradients lie in a predictable low-dimensional subspace which depends on the network architecture and incoming features. Exploiting this structure can significantly improve gradient-free optimization schemes based on directional derivatives, which have struggled to scale beyond small networks trained on toy datasets. We study how to narrow the gap in optimization performance between methods that calculate exact gradients and those that use directional derivatives. Furthermore, we highlight new challenges in overcoming the large gap between optimizing with exact gradients and guessing the gradients.

## 1 INTRODUCTION

Researchers have wondered for decades if neural networks can be optimized without using backpropagation to differentiate the loss function. One tantalizing possibility, first articulated by Polyak (1987) and Spall (1987), is to choose a *random direction* $y$, compute the *directional derivative* of the objective in the direction $y$, and take a step in the direction $y$ scaled by the directional derivative, $(y \cdot \nabla L)$. For a neural network with weights $w$, this step is an elegant unbiased estimator of the true gradient:

$$w_{t+1} = w_t - \alpha(y \cdot \nabla L)y \qquad (1)$$

This method has recently seen renewed interest because directional derivatives can be computed very efficiently using forward-more differentiation, which does not incur the immense memory cost of backpropagation (Chandra, 2021; Baydin et al., 2022; Silver et al., 2021).

Unfortunately, as the dimensionality of the optimization problem increases, so does the variance in the estimator, which in turn inhibits convergence. If the network has $N$ parameters, the cosine similarity between the guess and true gradient falls to $O(\frac{1}{\sqrt{N}})$. For neural networks with billions of parameters, this guess is nearly orthogonal to the true gradient and thus impractical.

For an $N$-dimensional problem, Nesterov & Spokoiny (2017) show that this method incurs an $O(N)$ slowdown over ordinary gradient descent. Belouze (2022) provides further theoretical and empirical evidence that the "curse of dimensionality" prevents this method from scaling. Similarly, Chandra (2021) can train a small MLP on MNIST but not a ResNet on the CIFAR-10 dataset. Work from Ren et al. (2022) also demonstrates this slowdown and proposes addressing it by including local loss functions after each block in the network.

Can we do any better by *guessing* more intelligently? At first glance, without auxiliary loss functions or access to labels, isotropically random guessing appears to be the best we can do. But the intrinsic dimension of gradients is often much lower than $N$ (Li et al., 2018), which suggests that it might be possible to do better. In this paper, we observe that the topology and activations of the neural network heavily constrain the gradient even before a loss is computed—that is, we know information about the gradient before we even see the label. Thus, we ask:

*How can we use knowledge about the network architecture and incoming features to make better gradient guesses?*

By carefully analyzing the structure of neural network gradients, we show it is possible to produce guesses with dramatically higher cosine similarity to the true gradient, even for networks with millions of parameters (Figure 1). We then compare these cosine similarities to those of stochastic

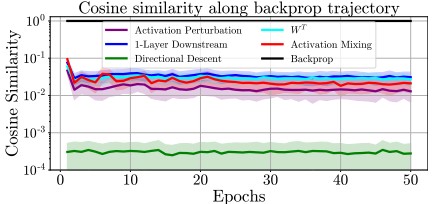

| Method | Cosine Similarity | 1-step effectiveness |
|---|---|---|
| Backprop (Oracle) | 1 | 1 |
| Directional Descent | $0.0003 \pm 0.00023$ | $1 \times 10^{-6} \pm 1 \times 10^{-6}$ |
| Activation Perturbation | $0.016 \pm 0.0083$ | $6.9 \times 10^{-4} \pm 8.4 \times 10^{-4}$ |
| Activation Mixing | $0.025 \pm 0.014$ | $3.4 \times 10^{-3} \pm 5.5 \times 10^{-3}$ |
| $W^T$ | $0.030 \pm 0.010$ | $1.7 \times 10^{-3} \pm 1.8 \times 10^{-3}$ |
| 1 Layer Downstream | $0.034 \pm 0.013$ | $2.7 \times 10^{-3} \pm 2.8 \times 10^{-3}$ |

Figure 1: **(left)** Guessed gradient cosine similarity for a 6-layer, 1024-wide MLP being trained on CIFAR10 using backpropagation. We track each method's cosine similarity along the backprop trajectory, and tabulate the average in the table on the right. Compared to directional descent, our proposed methods like $W^T$ achieve approximately $100\times$ larger average cosine similarity. **(right)** We also tabulate the average cosine similarity as well as the loss reduction for 1 step (relative to backprop). Our methods reduce the loss more than $1000\times$ more for a single batch.

gradient descent (SGD) to understand what cosine similarities are required for neural network optimization for MLPs (Table 2) and Ren et al. (2022) Mixer architecture (Table 1). Next, we analyze the variance and optimization properties of these guesses to highlight their improved convergence, and study limitations such as bias. Finally, we demonstrate an unexpected "*self-sharpening*" phenomenon, where the training dynamics induced by these guesses make it easier to guess the gradient over time. This phenomenon leads to $> 95\%$ training accuracy on CIFAR10 *without backpropagation*. Nonetheless, these advances come with some important limitations, which we also discuss — while the methods outlined in our work provide theoretical advances in our understanding of gradient structure, they are not yet ready for practical use. For example, they are still significantly slower than backpropagation with gradient checkpointing (Sohoni et al., 2019). Currently, all approaches in this space fall short of backpropagation-level performance on larger problems.

## 2 METHODS

In this section, we describe the proposed methods for narrowing the guess space. We begin by describing architecture-based constraints and then describe constraints based on knowledge about the relationship between gradients and activations. To facilitate further research, we will make the code available at the time of publication.

### 2.1 ARCHITECTURE-AWARE GRADIENT GUESSING

Suppose we optimize a $k$-layer MLP with weights $W_1, W_2, \ldots, W_k$ using ReLU activation functions. At some layer $i$, we take as input some incoming activations $x_i$, compute the "pre-activations" $s_i$, and then compute the "post-activations" $x_{i+1}$. We then pass $x_{i+1}$ onto the next layer, ultimately computing some loss $L$. Finally, we wish to compute or approximate $\partial L / \partial W_i$ to train that layer:

$$s_i = W_i x_i, \quad x_{i+1} = \text{ReLU}(s_i), \quad L = \ell(s_k) \tag{2}$$

The current state-of-the-art method is to "guess" the unknown $\partial L / \partial W_i$ uniformly at random via a spherically symmetric Gaussian. Can we do better?

By "unfolding" the computation, we can identify exploitable information to refine our gradient guesses. Considering the computation at layer $i$, where $s_i = W_i x_i$ represents the pre-activations, applying the chain rule reveals a crucial insight: $\partial L / \partial W_i$ is essentially the outer product of the (unknown) gradient at future layers and the (known) incoming activations $x_i$:

$$\frac{\partial L}{\partial W_i} = \frac{\partial L}{\partial s_i} \cdot \frac{\partial s_i}{\partial W_i} = \frac{\partial L}{\partial s_i} \cdot \frac{\partial W_i x_i}{\partial W_i} = \frac{\partial L}{\partial s_i} \cdot x_i^{\top} \tag{3}$$

Notice in particular that $\partial L / \partial s_i$ is of significantly lower dimension than $\partial L / \partial W_i$. This leads us to our **first insight**: we can "guess" the low-dimensional $\partial L / \partial s_i$ and use the known $x_i^{\top}$ to compute a much lower-variance guess for the high-dimensional $\partial L / \partial W_i$. Note that for a neural network with width $K$, each weight has $K \times K = K^2$ parameters, and we have reduced the guessing space from $O(K^2)$ to $O(K)$. Practically, for neural networks with millions of parameters, this means guessing

in a few thousand dimensions. This guess consists of perturbations of pre-activations ($s_i$), similar to the work of Ren et al. (2022) and we denote this as *activation perturbation*.

Let us keep unfolding. The next step is to take the ReLU of $s_i$ to obtain $x_{i+1}$.

$$\frac{\partial L}{\partial s_i} = \frac{\partial L}{\partial x_{i+1}} \cdot \frac{\partial x_{i+1}}{\partial s_i} = \frac{\partial L}{\partial x_{i+1}} \cdot \frac{\partial \text{ReLU}(s_i)}{\partial s_i} \tag{4}$$

Our **second insight** is that by the very nature of ReLU activations, the Jacobian matrix $\partial \text{ReLU}(s_i)/\partial s_i$ will be a sparse diagonal matrix. It is diagonal since each input controls one and only one output. Furthermore, this matrix will also typically "zero out" some entries of the incoming gradient. This suggests that we should "guess" only the surviving entries of $\partial L/\partial x_{i+1}$, as determined by that sparse and diagonal matrix (known at guess-time). This further decreases the dimensionality of our guessing space and, consequently, the variance of our guesses. Although the exact fraction depends on many factors, including the data and optimizer, the sparsity ratio is typically 0.5 at initialization. Let us unfold one last time, looking into the *next* weight matrix $W_{i+1}$. Again, we apply the chain rule, now at $s_{i+1}$:

$$\frac{\partial L}{\partial x_{i+1}} = \frac{\partial L}{\partial s_{i+1}} \cdot \frac{\partial s_{i+1}}{\partial x_{i+1}} = \frac{\partial L}{\partial s_{i+1}} \cdot \frac{\partial W_{i+1} x_{i+1}}{\partial x_{i+1}} = \frac{\partial L}{\partial s_{i+1}} \cdot W_{i+1} \tag{5}$$

As before, the future gradient $\partial L/\partial s_{i+1}$ is unknown and must be guessed. But we know that it will immediately be multiplied by $W_{i+1}^\top$. While this does not necessarily give a "hard" constraint on our guess, our **third insight** is that $W_{i+1}^\top$ often effectively has low rank (Huh et al., 2023). We can constrain our guesses to lie in the image of $W_{i+1}^\top$ by multiplying our guess with it to further lower the dimensionality of our guessing space. To summarize, we know that

$$\frac{\partial L}{\partial W_i} = \frac{\partial L}{\partial s_{i+1}} W_{(i+1)} \cdot \frac{\partial \text{ReLU}(s_i)}{\partial s_i} \cdot x_i^\top \tag{6}$$

At "guess time" all of these quantities are known except for $\partial L/\partial s_{i+1}$, which we guess as random normal with zero mean and unit variance. We then apply a series of constraints to mould it into a much more effective guess for $\partial L/\partial W_i$. We refer to the combination of these methods as "$W^\top$".
**Partial backpropagation**: The previous approach incorporates local architecture information into the gradient guess. As a more general approach, we can consider guessing the gradient for some neurons $x_{i+l}$ which are $l$ layers downstream of the current layer, and backpropagating through the intermediate portion of the graph.

$$\frac{\partial L}{\partial W_i} = \underbrace{\frac{\partial L}{\partial x_{i+l}}}_{\text{guess here}} \cdot \frac{\partial x_{i+l}}{\partial s_i} \cdot x_i^\top \tag{7}$$

This approach requires storing the intermediate activations for the $l$ layers, and in the full limit, is equivalent to regular backpropagation but with a random error vector. In our experiments, we find that $l > 1$ has diminishing returns, so we stick to $l = 1$. All aforementioned methods are special cases of this general approach.

## 2.2 FEATURE-AWARE GRADIENT GUESSING

We unroll SGD update steps and show that activations and gradients approximately lie in the same subspace. We visualize this phenomenon in Figure 2. The goal is to generate random vectors in the same subspace as the true gradient $\frac{\partial L}{\partial x_{k+1}}$. We use a random mixture of activations $x_{k+1}$ as the guess.

**Intuition**: Consider the downstream weight matrix $W_{k+1}$ being updated iteratively with SGD with a learning rate $\eta$. Then at timestep $t$:

$$W_{k+1}[t] = W_{k+1}[0] + \sum_{i=1}^{t-1} \Delta W_{k+1}[i] \tag{8}$$

$$= W_{k+1}[0] + \eta \sum_{i=1}^{t-1} \frac{\partial L}{\partial s_{k+1}}[i] x_{k+1}^T[i] \tag{9}$$

Figure 2: Activations and gradients approximately lie in the same subspace. For an MLP trained on MNIST digit classification, we plot (as images) for each class **(a)** the first principal component of gradients with respect to input images (top row), **(b)** the first principal components of the inputs (middle) and **(c)** random combinations of inputs (bottom row). Even though the MLP is initialized with random weights, and has no inductive bias towards images, the principal components of gradients look similar to inputs. Our "activation mixture" method uses random mixtures of activations to generate guesses in the same subspace as the gradients. **(Right)** Activation subspace is a much better match for gradients. We compute the PCA components of the activation subspace and compare it to a random subspace. We project the gradients onto these subspaces and measure the cosine similarity of the projection compared to the true gradient. We plot these curves for different widths, depths, layers, and epochs. Activation subspace consistently captures the gradient better.

and thus the term $\frac{\partial L}{\partial x_{k+1}}[t]$ can be expanded:

$$\frac{\partial L}{\partial x_{k+1}}[t] = \frac{\partial L}{\partial s_{k+1}}[t]W_{k+1}[t] \tag{10}$$

$$= \frac{\partial L}{\partial s_{k+1}}[t]W_{k+1}[0] + \eta \sum_{i=1}^{t-1} \beta_k[t,i]x_{k+1}^T[i] \tag{11}$$

Ignoring the first term (weight at initialization),

$$\frac{\partial L}{\partial x_{k+1}}[t] \approx \eta \sum_{i=1}^{t-1} \beta_k[t,i]x_{k+1}^T[i] \tag{12}$$

where $\beta_k[t,i] = \frac{\partial L}{\partial s_k}[t]^T \frac{\partial L}{\partial s_k}[i]$ measures the similarity of $s_k$ gradients at timesteps $t$ and $i$. We thus see that the desired gradient approximately lies in the subspace of previously generated activations $x_{k+1}[i]$. While this intuitive argument makes many assumptions (such as SGD optimizer and small weights at initialization), our experiments show that the activation subspace is often well-aligned with the gradient subspace across depths, widths, and training epochs (Figure 2).

We use this observation to generate a guess for $\frac{\partial L}{\partial x_{k+1}}$ by taking current activations $x_{k+1}$ and computing random linear combinations of all the training example activations in the batch. We call this method "activation mixing".

In summary, we propose 4 variants: (1) **_Activation perturbation_**, which produces isotropic guesses in activation space rather than weight space; (2) **_Activation mixing_**, which uses mixtures of activations as the guess; (3) $W^\top$, which multiplies an isotropic guess by the transpose of the weight matrix $W$ to produce a guess; and (4) 1-**_layer downstream_**, which backpropagates a guess from the layer next to the current layer.

## 3    RESULTS

We evaluate the directional descent baseline, activation perturbation baseline, activation mixing, $W^\top$, and "1-Layer Downstream". We compare each method's cosine similarity and optimization performance. We further analyze phenomena like bias. Please see the supplementary material (A.2) for implementation details.

**Cosine similarity along backprop**: To compare each method's gradient guess, we start with an MLP with depth 6 and width 1024. This MLP is trained for 50 epochs on the CIFAR10 dataset using a batch size of 512 and learning rate of $10^{-4}$ using the AdamW optimizer. After each epoch, we

measure the cosine similarities of guessed gradients for each method and plot the resulting curves in Figure 1. We also compute the average and standard deviation along this path and tabulate it. We find that, on average, our proposed methods such as $W^\top$ produce cosine similarities that are hundreds of times higher than directional descent. **One step effectiveness**: Cosine similarity only measures the effectiveness of a method in the infinitesimal limit of the step size. It ignores the curvature of the loss landscape, and thus can be a misleading measure of a method's effectiveness. We directly compare each method's effectiveness (loss reduction relative to backprop) and further search over multiple step sizes $[10^{-5}, 10^{-4}, 10^{-3}, 10^{-2}, 10^{-1}]$ for each method (and backprop) to make the comparison as fair as possible. We find that our methods are thousands of times more effective compared to directional descent in the 1-step regime.

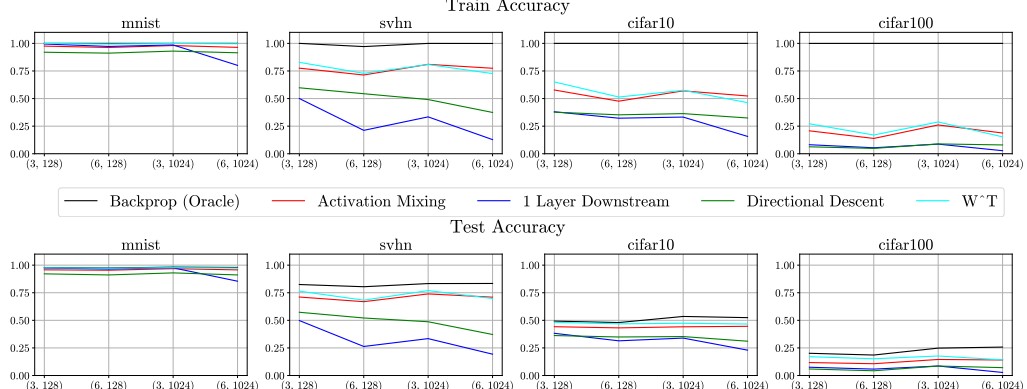

Figure 3: Our proposed methods outperform directional descent. We plot MLP train and test accuracies for various methods and datasets. Top row consists of train accuracy plots and bottom row consists of test accuracy plots. The columns refer to MNIST, SVHN, CIFAR10, and CIFAR100 respectively. The x-axis in each plot is labelled as (depth, width) for each MLP configuration, and sorted by the number of parameters. We see that for every dataset, our proposed methods achieve significantly higher accuracy than directional descent. The gap between our methods and backprop training accuracy increases with dataset complexity (e.g. CIFAR100 vs. MNIST), whereas test accuracy is more comparable. Please refer to Table 2 for details.

**Training MLPs on MNIST, SVHN, CIFAR10, CIFAR100**: We next conducted experiments to train MLPs using our proposed methods on four commonly used datasets: MNIST, SVHN, CIFAR10, and CIFAR100. The MLPs were configured with four (depth, width) configurations: (3, 128), (6, 128), (3, 1024), (6, 1024). These settings were chosen to evaluate the effect of depth and width on the learned network accuracy. Since our proposed methods can be significantly slower than backpropagation, each method was trained for 1000 epochs. We used the same batch size of 512, learning rate of $10^{-4}$ and AdamW optimizer. The resulting train and test accuracies are reported in Table 2, and the plots are reported in Figure 3. While our proposed methods outperform directional descent, there is a large gap between these methods and backprop, and the gap grows larger with more complex datasets. In the next few experiments, we explore some possible reasons for this gap.

**Comparison against Ren et al. (2022)**: To test our method's effectiveness for large-scale models, we evaluate our model on the Local Mixer architecture proposed in Ren et al. (2022) (Table 1). We also use the Adam optimizer (LR=$10^{-3}$) and image augmentations to extract as much accuracy from the model as possible. Adam significantly boosts the model accuracy for all baselines and our methods, and the same is true for image augmentations (random cropping, random horizontal flips). These two changes are sufficient for a $9.8\%$ increase in the baseline accuracy. In the Adam setting, our method achieves a $1.3\%$ gain on top of the method described in Ren et al. (2022). A similar gap persists as we move to the augmentation setting. Since augmentations slow down convergence, we let all the non-backprop methods train for $10\times$ longer. In that setting, our method again achieves the highest accuracy ($77.4\%$), beating backprop by $1\%$ and Ren et al. (2022) method by $1.4\%$. We find that in contrast to our MLP experiments, our method actually generalizes *better* than backprop on the Local mixer architecture. We further analyze these results in Appendix A.4 and hope to study this finding in detail in future work.

|  | Backprop | Ren et al. (2022) | Mixing | $W^T$ |
|---|---|---|---|---|
| Reported (Ren et al. (2022)) | 66.4 | 69.3 | - | - |
| Reproduced with Adam | 71.2 | 71.2 | 68.8 | **72.5** (+1.3) |
| Augmentation (500 epochs) | **76.4** | 72.2 | 68.2 | 74.4 |
| Augmentation (5000 epochs) | - | 76 | 69.4 | **77.4** (+1.4) |

Table 1: Test accuracies for our methods as well as baselines on the Local Mixer architecture from Ren et al. (2022) on CIFAR10. Our method $W^T$ achieves higher test accuracy than backpropagation and activation perturbation. Adding augmentations and using Adam boosts this accuracy significantly compared to the reported baselines.

**Effect of bias on our methods**: We measure the behavior of our methods in the limit of a large number of guesses. For an unbiased method such as directional descent or activation perturbation, more guesses result in a better cosine similarity (increasing proportionally to $O(\sqrt{G})$ for $G$ guesses). This is not the case for biased methods, as decreased variance is traded off with increased bias. We pick an MLP with depth 6 and width 1024 and train it on CIFAR10 for 1 epoch to represent a neural network during its training process. We then pick a single example and sample $2, 4, 8, \ldots, 4096, 8192$ guesses for each method, plotting the resulting cosine similarities in Figure 4. We find that methods such as $W^\top$ and activation mixing saturate in cosine similarity after approximately 1000 guesses, whereas the unbiased "activation perturbation" baseline improves consistently with more samples. This difference highlights the bias present in our proposed methods. This bias is further analyzed in the next experiments, Section 4 and Appendix A.3.

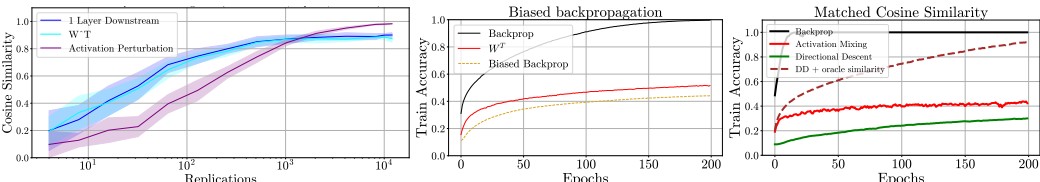

Figure 4: **(left)** Our methods (Mixing, Downstream, $W^T$) are biased estimators of the gradient. For a single example input, we average the multiple guesses and plot cosine similarity as a function of the number of guesses. In contrast to an unbiased random baseline where averaging over more guesses leads to better cosine similarities, the cosine similarity quickly saturates for the biased methods. **(middle)** Adding the bias of $W^T$ to backprop makes it fail in the same way. We add the bias from $W^T$ to backprop and find that it saturates at a similar training accuracy, indicating that the bias is sufficient to hamper optimization. **(right)** The cosine similarity achieved by our methods, without the bias, is sufficient to achieve high training accuracy on tasks like CIFAR10

**Effect of bias on the correct guess**: To understand the effect of bias in isolation from the variance caused by guessing, we apply the bias of $W^\top$ to the actual gradient calculated with backpropagation ($\hat{g} = W^\top W g$, explained further in Section 4) and plot the results in Figure 4. The resulting algorithm, despite less variance, fails to converge as quickly as the unbiased backpropagation and has a training curve similar to $W^\top$, indicating that the bias alone is sufficient to hamper optimization.

**Effect of bias vs. variance**: To better understand the effect of bias and low/high cosine similarity on the network training accuracy, we construct a version of directional descent where we artificially modify its guess to have similar cosine similarity to $W^\top$. We use the $(6, 1024)$ MLP, where the cosine similarity for $W^\top$ is approximately $0.03$. We ensure that the modified directional descent guess always has a cosine similarity of $0.03$ with the true gradient. We do this by decomposing the original guess into the components along the gradient and perpendicular to the gradient, normalizing each part, and spherically interpolating them. Please refer to the supplementary material (A.2) for implementation details. We compare this modified directional descent to $W^\top$ and the original directional descent in Figure 4. While $W^\top$ converges significantly faster than the original directional descent, its bias hampers its convergence speed, especially when compared to the modified directional descent. This experiment also highlights that a cosine similarity of $0.02$ is sufficient to reach high training accuracy for datasets such as CIFAR10, and bias is the key limiting factor.

**Gradients and activations approximately lie in the same subspace**: We compute the cosine similarity between the true gradient and its projection onto the subspace spanned by activations. If the activation and gradient subspaces are approximately aligned, the cosine similarity between the gradient and its projection should be high. We pick the basis for the activation subspace by running PCA and using the principal components as the basis vectors. We contrast this to a random subspace created by randomly sampling a set of vectors and orthonormalizing them. We plot the resulting curves for each layer in MLPs of depth 3,4, or 6, widths 1024, and during all 20 training epochs. We see that the activation subspace consistently requires much fewer basis vectors for a significantly better approximation than a random subspace, getting cosine similarity as high as 0.5 with less than 10 principal components (in contrast, random subspace gets 0.1 cosine similarity).

## 3.1 THE SELF-SHARPENING PHENOMENON

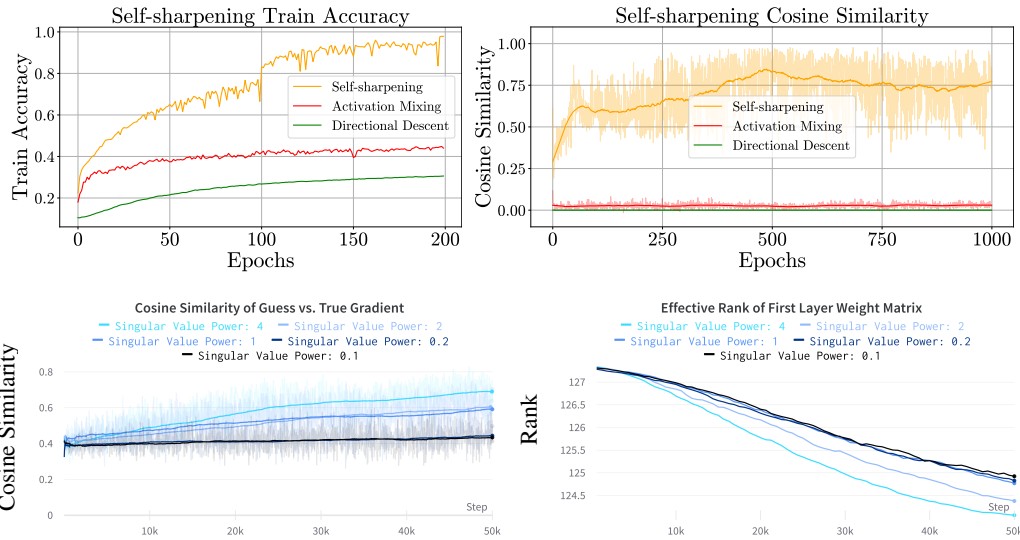

Figure 5: **(Top row)** Self-sharpening effect leads to the cosine similarity increasing over the course of training and a higher training accuracy as a result. **(Bottom row)** We can re-create this effect by manipulating the singular values for $W^T$. By raising the singular values to different powers, we can make some singular values dominate the guess. This leads to the weight matrices becoming lower rank over time, and thus higher cosine similarity. The gradients become easier to guess.

We report a peculiar "self-sharpening" behavior seen in some methods, the space of guesses becomes more narrow or 'sharpens' over the course of training, improving the cosine similarity to the exact gradient. As seen in Figure 5, the cosine similarity increases sharply compared to other methods, such as Directional descent, where it stays nearly constant. While we do not know the precise cause of this effect, we hypothesize that this effect happens due to a feedback loop of decreasing rank of downstream weights. This decreasing rank narrows the guess space, which makes updates less diverse and further lowers the rank.

To this end, we design a gradient guessing scheme with these qualities. We use random uniform noise in the "1-layer downstream" gradient guess, and to further speed up the convergence, we replace the last layer's guess with the true error vector (since it is local information and does not require any backpropagation). Please refer to the supplementary section (A.2) for experimental. This change drastically increases the cosine similarity to values as high as 0.6 over time. As a result, the training accuracy also reaches $> 95\%$ on CIFAR10. However, while this phenomenon achieves high training accuracy, it also hurts generalization, reaching only 33% test accuracy.

**Replicating self-sharpening by manipulating singular values**: We hypothesize that the biased guesses dominated by a few singular values lead to lower rank weight matrices, which lead to higher cosine similarity over the course of training. Here, we modify the $W^\top$ guessing method by computing the singular value decomposition of W and raising its singular values to various powers [0.1, 0.2 ..., 4]. Higher powers lead to a more imbalanced distribution, with a few singular values dominating

the rest, whereas smaller powers lead to all singular values becoming nearly equal. We plot the resulting cosine similarity and effective rank (Roy & Vetterli, 2007) in Figure 5.

We see that when a few singular values dominate, it leads to a lower weight matrix rank and higher cosine similarity, which increases over the duration of training in lockstep with the lowering rank. Conversely, the lowest powers lead to the smallest increase. The first plot shows the cosine similarity of our guess vs. the true gradient. The increasing cosine similarity demonstrates the self-sharpening effect. The second plot plots the effective rank (Roy & Vetterli, 2007) for the weight matrix corresponding to the first layer of the network. Effective rank is a continuous approximation of the matrix rank and is commonly used when many singular values are close to zero.

## 4 ANALYSIS

We discuss one possible source of the bias present in our proposed methods. We start with the unbiased estimator which uses the Jacobian Vector Product (JVP):

$$\hat{g} = (\nabla L.y)y \tag{13}$$

and we compute the expectation of this estimator:

$$\mathbb{E}\big[\hat{g}\big] = \mathbb{E}\big[(\nabla L.y)y\big] = \mathbb{E}\big[yy^T\big]\nabla L = \text{Cov}(y)\nabla L \tag{14}$$

thus, in expectation, the gradient guess is equal to the original guess scaled by the covariance matrix of the guess. Thus the bias is:

$$\mathbb{E}\big[\hat{g}\big] - \nabla L = (\text{Cov}(y) - I)\nabla L \tag{15}$$

Therefore, the guess can only be unbiased if the covariance matrix is equal to the identity matrix in the subspace that the gradients lie in.

For our proposed estimators, this is easily shown to be false. Activation mixing uses random mixtures of activations as the gradient guesses, and thus its covariance matrix is the same as the covariance matrix of the activations (and thus non-identity). Our methods rely on these covariance matrices being low rank and well-aligned with the gradient subspace to produce high cosine similarity guesses. Still, as a result, our expected guess is also scaled by these covariance matrices, thus biased. In future work, we hope to use this information to undo the bias caused by these non-identity covariance matrices.

**Bias for** $W^\top$: The $W^\top$ method involves sampling a random normal noise vector and transforming it with $W^\top$ to confine it to the range space. Thus, the guess $y$ for any given layer can be written as:

$$y = W^\top \epsilon$$

where $\epsilon$ is a random normal vector, i.e., $\epsilon_i \sim \mathcal{N}(0, 1)$. Thus the guess vector $y$ is also a multivariate normal vector with the covariance matrix:

$$\text{Cov}(y) = W^\top W$$

and so the bias for each layer's activation is given by:

$$\implies \text{Bias}[W^\top] = (W^\top W - I)\nabla L$$

**Why more layers are not always better**: Why not use as many steps as possible in partial backpropagation (e.g., using the next 2 or 3 downstream layers)? In practice, the bias can increase with each additional layer. Here, we show how with a simple toy example.

Let the activation vector at Layer $i$ be represented by a vector $x_i \in \mathbb{R}^n$, and let the jacobians of the next few layers (i.e. layers $i + 1, i + 2, \ldots, i + k$) be represented by $J_{i+1}, J_{i+2}, \ldots J_{i+k} \in \mathbb{R}^{n \times n}$ (here we assume that all layers are the same width without loss of generality). We denote their product, the accumulated jacobian for layers $i + 1$ to $i + k$, as $J$ for notational simplicity. Also, let $g_k \in \mathbb{R}$ be the true corresponding gradient.

We begin by noting that $g_i = J\frac{\partial L}{\partial x_{i+k}}$ by chain rule. Thus, $g_i$ lies in the range space of $J$, i.e. $g_i \in \mathcal{R}(J)$. Using this knowledge can significantly reduce the guessing space.

**Train Accuracy**

| | Method | CIFAR100 | CIFAR10 | SVHN | MNIST |
|---|---|---|---|---|---|
| D:3, W:128 | Directional gradients | 6.7 ± 0.3 | 37.4 ± 0.2 | 59.8 ± 0.4 | 92.3 ± 0.2 |
| | Activation Perturbation | 13.3 ± 0.1 | 45.9 ± 0.3 | 69.3 ± 1.3 | 98.7 ± 0.1 |
| | Mixing (ours) | 20.1 ± 0.6 | 57.2 ± 0.8 | 76.34 ± 0.6 | 96.9 ± 0.5 |
| | $W^T$ (ours) | 25.6 ± 0.2 | 62.4 ± 0.8 | 81.3 ± 0.9 | 100.0 ± 0 |
| | Downstream (ours) | 30.8 ± 0.7 | 65.6 ± 0.5 | 83.2 ± 0.8 | 99.9 ± 0.1 |
| | Self-sharpening (ours) | 11.4 ±1.7 | 51.5 ±5.5 | 71.6 ±1.9 | 99.5 ±0.3 |
| | Backprop (oracle) | 100.0 ± 0 | 100.0 ± 0 | 99.5 ± 0.9 | 100.0 ± 0 |
| D:6, W:128 | Directional gradients | 5.2 ± 0.4 | 35.9 ± 0.2 | 54.8 ± 1.2 | 91.4 ± 0.2 |
| | Activation Perturbation | 9.4 ± 0.5 | 38.9 ± 0.4 | 56.1 ± 1.8 | 97.3 ± 0.3 |
| | Mixing (ours) | 13.1 ± 0.7 | 48.5 ± 1.3 | 72.1 ± 1.3 | 96.6 ± 0.3 |
| | $W^T$ (ours) | 16.5 ± 0.4 | 51.5 ± 0.2 | 73.2 ± 1.1 | 99.2 ± 0.1 |
| | Downstream (ours) | 16.1 ± 0.4 | 52.8 ± 0.8 | 77.0 ± 1.2 | 99.5 ± 0.1 |
| | Self-sharpening (ours) | 21.1 ±8.5 | 42.5 ±15.4 | 78.9 ±11.2 | 100.0 ±0.0 |
| | Backprop (oracle) | 100.0 ± 0 | 100.0 ± 0 | 100.0 ± 0 | 100.0 ± 0 |
| D:3, W:1024 | Directional gradients | 9.0 ± 0.1 | 35.9 ± 0.2 | 54.8 ± 1.2 | 91.4 ± 0.2 |
| | Activation Perturbation | 9.9 ± 0.6 | 36.7 ± 2.7 | 50.1 ± 5.6 | 96.2 ± 1.2 |
| | Mixing (ours) | 26.9 ± 0.4 | 59.4 ± 1.2 | 80.8 ± 1.8 | 98.4 ± 1.2 |
| | $W^T$ (ours) | 28.2 ± 0.5 | 56.9 ± 1.4 | 79.8 ± 1.3 | 99.6 ± 0.3 |
| | Downstream (ours) | 28.4 ± 0.7 | 58.5 ± 1.0 | 80.6 ± 0.5 | 99.9 ± 0.1 |
| | Self-sharpening (ours) | 23.1 ±1.1 | 70.8 ±15.0 | 69.8 ±5.2 | 100.0 ±0.0 |
| | Backprop (oracle) | 100.0 ± 0 | 100.0 ± 0 | 100.0 ± 0 | 100.0 ± 0 |
| D:6, W:1024 | Directional gradients | 7.8 ± 0.1 | 32.2 ± 0.1 | 37.4 ± 0.2 | 91.4 ± 0.3 |
| | Activation Perturbation | 3.6 ± 0.4 | 26.9 ± 1.6 | 24.3 ± 4.5 | 95.5 ± 0.8 |
| | Mixing (ours) | 18.1 ± 0.4 | 52.2 ± 1.3 | 78.4 ± 2.2 | 97.2 ± 0.2 |
| | $W^T$ (ours) | 14.6 ± 0.3 | 46.7 ± 0.9 | 73.6 ± 0.8 | 99.4 ± 0.2 |
| | Downstream (ours) | 9.7 ± 0.3 | 37.6 ± 2.6 | 71.3 ± 0.8 | 99.4 ± 0.2 |
| | Self-sharpening (ours) | 18.1 ±9.1 | 97.5 ±3.9 | 97.9 ±0.5 | 99.9 ±0.1 |
| | Backprop (oracle) | 99.9 ± 0.1 | 100.0 ± 0 | 100.0 ± 0 | 100.0 ± 0 |

**Test Accuracy**

| | Method | CIFAR100 | CIFAR10 | SVHN | MNIST |
|---|---|---|---|---|---|
| D:3, W:128 | Directional gradients | 6.1 ± 0.4 | 35.8 ± 0.4 | 57.4 ± 0.4 | 92.2 ± 0.2 |
| | Activation Perturbation | 12.2 ± 0.2 | 42.3 ± 0.5 | 65.4 ± 1.3 | 96.9 ± 0.2 |
| | Mixing (ours) | 11.6 ± 0.3 | 44.2 ± 0.5 | 69.6 ± 0.4 | 95.7 ± 0.2 |
| | $W^T$ (ours) | 17.1 ± 0.3 | 46.9 ± 0.8 | 76.2 ± 0.4 | 97.5 ± 0.1 |
| | Downstream (ours) | 18.1 ± 0.3 | 47.8 ± 0.7 | 76.9 ± 0.4 | 97.5 ± 0.1 |
| | Self-sharpening (ours) | 10.6 ±0.9 | 35.4 ±2.1 | 64.8 ±1.6 | 94.2 ±0.5 |
| | Backprop (oracle) | 19.7 ± 0.1 | 49.0 ± 0.3 | 81.8 ± 0.2 | 97.8 ± 0.1 |
| D:6, W:128 | Directional gradients | 4.6 ± 0.4 | 34.4 ± 0.5 | 53.2 ± 1.3 | 91.4 ± 0.1 |
| | Activation Perturbation | 9.9 ± 0.5 | 38.9 ± 0.3 | 53.4 ± 1.1 | 96.2 ± 0.1 |
| | Mixing (ours) | 10.6 ± 0.5 | 42.5 ± 1.0 | 66.4 ± 0.9 | 95.3 ± 0.2 |
| | $W^T$ (ours) | 14.1 ± 0.3 | 46.5 ± 0.4 | 69.4 ± 1.6 | 96.9 ± 0.2 |
| | Downstream (ours) | 14.1 ± 0.3 | 46.5 ± 0.5 | 72.8 ± 1.0 | 97.0 ± 0.1 |
| | Self-sharpening (ours) | 14.0 ±2.7 | 43.2 ±1.1 | 70.6 ±0.8 | 96.2 ±0.2 |
| | Backprop (oracle) | 17.9 ± 0.4 | 47.5 ± 0.3 | 80.2 ± 0.3 | 97.4 ± 0.1 |
| D:3, W:1024 | Directional gradients | 8.0 ± 0.2 | 34.8 ± 0.3 | 47.2 ± 1.0 | 92.8 ± 0.1 |
| | Activation Perturbation | 9.4 ± 0.2 | 38.3 ± 0.4 | 55.6 ± 1.1 | 96.6 ± 0.2 |
| | Mixing (ours) | 15.0 ± 0.3 | 44.6 ± 0.7 | 73.3 ± 0.9 | 97.0 ± 0.2 |
| | $W^T$ (ours) | 17.6 ± 0.5 | 48.0 ± 0.4 | 75.3 ± 0.6 | 97.7 ± 0.1 |
| | Downstream (ours) | 18.3 ± 0.3 | 48.4 ± 0.5 | 76.4 ± 0.5 | 97.8 ± 0.0 |
| | Self-sharpening (ours) | 11.9 ±1.4 | 38.7 ±1.1 | 65.1 ±4.0 | 96.1 ±0.1 |
| | Backprop (oracle) | 25.0 ± 0.5 | 53.8 ± 0.1 | 83.2 ± 0.1 | 98.3 ± 0.1 |
| D:6, W:1024 | Directional gradients | 6.9 ± 0.5 | 31.7 ± 0.3 | 37.8 ± 0.4 | 91.3 ± 0.3 |
| | Activation Perturbation | 4.7 ± 0.2 | 32.4 ± 0.6 | 31.8 ± 2.0 | 96.0 ± 0.1 |
| | Mixing (ours) | 14.0 ± 0.2 | 44.9 ± 0.5 | 72.5 ± 0.8 | 95.7 ± 0.2 |
| | $W^T$ (ours) | 14.0 ± 0.5 | 45.2 ± 0.5 | 70.7 ± 0.6 | 97.4 ± 0.1 |
| | Downstream (ours) | 10.2 ± 0.4 | 40.6 ± 0.4 | 69.6 ± 1.3 | 97.5 ± 0.1 |
| | Self-sharpening (ours) | 10.1 ±0.5 | 33.5 ±2.9 | 68.1 ±0.8 | 93.7 ±1.9 |
| | Backprop (oracle) | 25.2 ± 0.3 | 52.4 ± 0.3 | 82.9 ± 0.1 | 97.9 ± 0.2 |

Table 2: Train and test accuracies for all our proposed methods as well as the self-sharpening effect. We note that while the self-sharpening effect can deliver high train accuracy, it prevents the model from generalizing and thus achieving high test accuracy.

Let's look at how our methods use this knowledge: They sample a random normal vector $n_i \in \mathbb{R}^\ltimes$ multiply it by the jacobian to generate the guess direction $y = Jn$ for some normal vector $n \in \mathbb{R}^n$. This guess $y$ is used in a forward JVP to generate an estimate $\hat{g} = JVP(y)y = (g.y)y$

Using equation 15, the bias of $\hat{g}$ is:

$$\mathbb{E}[\hat{g}] - g = (Cov(y) - I)g = (Cov(Jn) - I)g = (JJ^T - I)g \tag{16}$$

The method predicts $\hat{g} = JJ^T g$ instead of the true $g$, resulting in the bias $(JJ^T - I)g$. To show this bias can increase with more layers, we consider a simple case where each $J_i = 2 * I$. Then $J = J_{i+1} \ldots J_i + k = 2^k I$, and the bias is $(4^k - 1)||g||$, which increases with $k$.

## 5  SUMMARY AND DISCUSSION

We show it is possible to produce gradient guesses with dramatically higher cosine similarities than directional descent. We then study the optimization properties of these guesses and highlight their improved convergence and limitations like bias. We show that bias is a major limiting factor for the scalability of our methods. Finally, we show the self-sharpening phenomenon, which helps us achieve $> 95\%$ training accuracy on CIFAR10 without backpropagation but also generalizes poorly. These findings not only suggest the potential of exploiting structure in gradients, but they also demonstrate new phenomena that could be a potential future research direction.

Since the bias is a major limitation, fixing this problem could unlock significant progress in scaling these methods to large-scale problems. This may be especially impactful for training models with model parallelism, and a better understanding of gradients may be useful for building more efficient optimization pipelines. Effective gradient guessing methods can also reduce memory consumption and help larger models fit on consumer-grade hardware, which can help democratize access to training, fine-tuning, and personalizing deep learning models.

Another interesting future research direction is applications in biologically plausible learning algorithms. Scalable credit assignment with biological constraints is an open problem, and many solutions include some variation of random perturbations (Jiang et al., 2023; Salimans et al., 2017; Hinton, 2022). Our proposed methods could be combined with recent data-based gradient guessing schemes Fournier et al. (2023). Heuristics such as the alignment of activation and gradient subspaces may be applied to associative memories in general and could prove useful for narrowing the guess space in such settings.

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

# A SUPPLEMENTARY MATERIALS

In this section: We run a hyperparameter sweep over learning rates and optimizers to check for the convergence speed of each method (A.1). We also describe experimental details for the experiments mentioned in the paper (A.2). We analyze bias further and demonstrate the claim that more downstream layers don't help (A.3). We also show further analysis for the Mixer experiments (A.4). We show applications of this work in fine-tuning models (A.5). We also plot activation subspaces (A.6) and the training curves for MLP experiments (A.7). All our code is implemented in PyTorch (Paszke et al., 2019).

## A.1 LEARNING RATE AND OPTIMIZER SWEEP

To test each method's convergence speed, we run a sweep over learning rates $[10^{-2}, 10^{-3}, 10^{-4}, 10^{-5}]$ and optimizers [AdamW (Loshchilov & Hutter, 2017), SGD, StableAdamW (Wortsman et al., 2023)]. We fix the same MLP architecture (3 layers, width 128) and dataset (CIFAR10). We plot the training accuracy for each method and each optimizer, choosing the best learning rate (defined as the highest train accuracy at the end of 1000 epochs) (Figure 6). We see that backprop predictably converges much faster than other methods, and gradient guessing methods such as directional descent and ours benefit greatly from Adam.

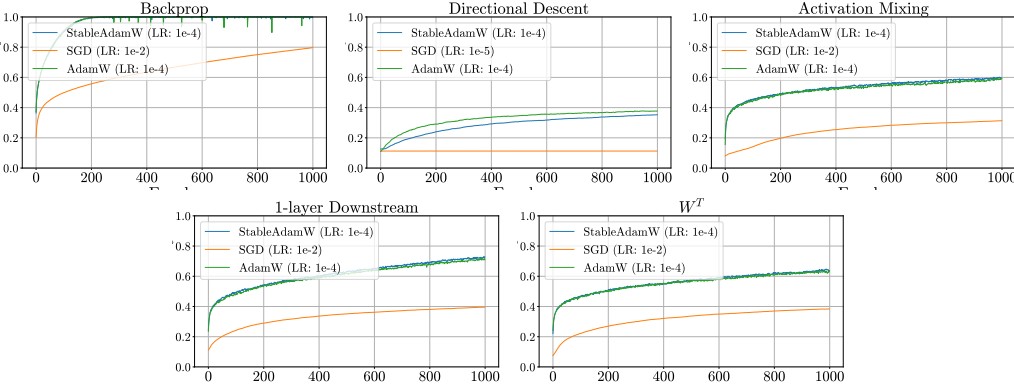

Figure 6: We plot each method's training curves for different optimizers on CIFAR10 for an MLP with 3 layers and width 128. We note that backprop converges significantly faster than directional descent and our proposed method. We also note that directional descent and our methods benefit greatly from variants of the Adam optimizer.

## A.2 EXPERIMENTAL DETAILS

**Jacobian-vector product (JVP) and forward gradients**: JVP takes as input a function $f(I)$, inputs $i$ (called primals), and perturbations $p$ (called tangents), and computes $J_f(I)|_{I=i} \cdot p$, where $J_f$ refers to the jacobian of $f$ and $J_f \cdot p$ refers to multiplying the jacobian matrix $J_f$ with the perturbations $p$. It measures the effect of infinitesimal perturbations $p$ around the original inputs $i$. This can be used for gradient estimation in two ways: (a) weight perturbations and (b) pre-activation perturbations.

For weight perturbations, the process is the same as used in Baydin et al. (2022). A gradient estimate $\hat{g}$ of the true gradient $g = \nabla_W L$ is generated (for directional descent, this is a random normal vector). Given such a guess, the JVP $(g.\hat{g})$ is computed. Note that this process does not require knowledge of the true gradient or any backward pass. The final update is just the guess scaled by the JVP, $\hat{g} \cdot \text{JVP} = \hat{g} \cdot (\hat{g} \cdot g)$

For pre-activation perturbations, the guesses and gradients are for pre-activations (i.e. the neuron values after a linear layer and before an activation function like ReLU). The new guess $\hat{g} = \nabla_{\vec{x}} L$ perturbs each pre-activation, and the JVP of this perturbation is measured. After scaling thiw pre-activation guess with the JVP, it can be converted into a weight update by computing the outer product as normally done in backpropagation $\Delta W_k = x_k g_k^T$.

Note that unlike weight perturbations, pre-activation perturbations apply separately for each batch element. Thus the JVP is measured separately for each measurement as well (i.e., we get as many JVP values as the batch size instead of one JVP value per batch). This allows pre-activation perturbations to take advantage of batch parallelism and get more gradient information for the same computation for each batch.

**Directional descent baseline**: As described above, our guess is a random normal vector. Each entry is given by $\hat{g}_i \sim \mathcal{N}(0, \frac{1}{\sqrt{N}})$. The division by $\sqrt{N}$ ensures that the norm of the guess doesn't grow with the parameter count.

**Activation perturbation baseline**: This baseline uses random normal perturbations for activations instead of weights. As described above, after scaling the guess with the JVP, we convert the scaled pre-activation guess to a weight update using the outer product.

In practice, this conversion is implemented using the ".backward()" function available for linear layers in PyTorch. We note that although we use ".backward()" to update individual layers one at a time, we are not using backpropagation to generate the guesses or to evaluate the JVP. This process can be done in a second forward pass once the JVP has been computed and does not require storing activations for all layers, unlike backpropagation.

**Activation mixing**: For each layer $k$, we take the batch of activations $\big[x_k[1], \ldots, x_k[B]\big]$ where $B$ is the number of batch elements. We compute a random linear combination $\alpha_1 x_k[1] + \ldots + \alpha_B x_k[B]$, where each weight $\alpha_i \sim \mathcal{N}(0, 1)$. To incorporate sparsity, we multiply each resulting guess by the ReLU activation mask $\frac{\partial \text{ReLU}(x_k[i])}{\partial x_k[i]}$. We also normalize the overall guess to ensure stability during training. In practice, this is implemented using backprop, as mentioned above. Since activation mixing does not apply to the last layer, we use a random normal guess for the last layer.

$\mathbf{W}^T$: For each batch element, we start with a random normal vector and multiply it by the transpose of the next layer's weight matrix $W_{k+1}^T$. We incorporate sparsity by multiplying with the ReLU mask as mentioned above and create a weight update in a similar fashion to previous methods. Similar to activation mixing, the last layer receives a random normal guess.

**1-layer downstream**: For each batch element, we trace the intermediate activations until the next layer and backpropagate a random normal vector through this graph. This method backpropagates signals from one layer to another but not across multiple layers as in regular backpropagation. We also note that the backpropagated vector is random and thus contains no information about the true gradient other than being in the same subspace. The update method and last layer treatment are the same as previous methods.

**Self-sharpening experiments**: For the self-sharpening experiments, we use the same framework as 1-layer downstream, but we use random uniform guessing instead of random normal, and the last layer gradients are replaced with the true error vector instead of a random guess. The last layer gradients are not required to see this effect, but they drastically speed up convergence. Since this method can be unstable, we also use the StableAdamW optimizer. **Self-sharpening through singular value manipulation**: This experiment was conducted on an MLP with 6 layers and 128 units per layer. We trained the MLP on CIFAR10 with batch size 128, LR $10^{-4}$ 0 weight decay, and 64 replicates for 50000 iterations.

**Artificially modifying directional descent's cosine similarity**: For our bias experiments, we modify directional descent to see how it would perform with a cosine similarity like our methods. To achieve this modified directional descent algorithm, we blend the random guess with the true gradient computed using full backpropagation. Specifically, we compute the two components of the guess: one along the gradient ($g_{\|}$) and one orthogonal to it ($g_{\perp}$). These two components are normalized to create an orthogonal basis. Then, a guess that has an angle $\theta$ with the gradient can be created as: $\hat{g}_\theta = \cos\theta g_{\|} + \sin\theta g_{\perp}$

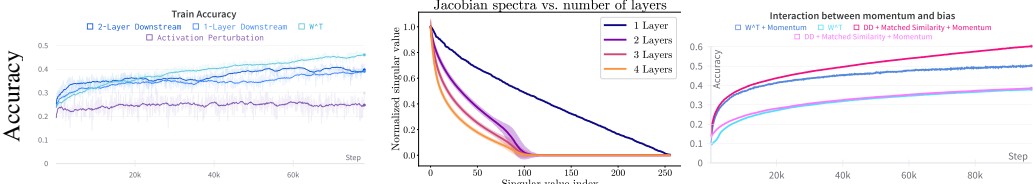

Figure 7: **(Left)** Additional downstream layers don't help. Using various methods, we train a 6-layer, 1024-wide MLP on CIFAR10 and plot their training curves above. We see that 2-layer downstream has no advantage over 1-layer downstream, indicating that the increase in bias counteracts any variance benefits. **(Middle)** Jacobian spectra for a different number of layers. The Jacobian for more layers has a faster-decaying spectrum and thus a lower effective rank, which makes it easier to guess the gradient. This can also increase the bias significantly and cause diminishing returns. **(Right)** Momentum and bias interact. We replicate the matched cosine similarity analysis from Figure 4 (right), but using SGD with/without momentum. We find that when momentum is used, bias causes a large slowdown for our method related to the unbiased version (as observed in the paper). However, without momentum, the difference between the two conditions is negligible. This may indicate a crucial role of momentum in this gradient guessing problem.

### A.3 FURTHER BIAS ANALYSIS

**Additional downstream layers don't help**: For the $6 \times 1024$ MLP setting on CIFAR, we train multiple models, each with different levels of partial backpropagation. We found that the increasing bias from including more layers results in diminishing returns, and additional layers in the partial backpropagation chain do not seem to speed up convergence or increase training accuracy (Figure 7 (left)).

**Multiple layers and jacobian spectra**: To show how multiple layers of a neural network can narrow the guess space (as well as increase bias), we compute the jacobian matrix for 1, 2, 3, and 4 layers for a 6x256 MLP. We find that with each additional layer, the jacobian spectrum decays faster, thus indicating a narrower effective guessing space (as well as more bias) (Figure 7 (middle)).

**Role of momentum in hindering biased estimators**: In this experiment, we explore the interaction between bias and optimization dynamics, specifically momentum. We compare a biased estimator ($W^\top$) to its cosine-similarity-matched unbiased version as in Figure 4(right). We find that in the presence of momentum, these two diverge from each other, as shown previously. However, the two training curves no longer diverge when momentum is removed (Figure 7 (right)). These results could indicate that the bias effects seen in this paper may be dependent on optimization dynamics.

Figure 8: Train and test accuracy curves for the Mixer experiments with Ren et al. (2022) architecture and losses. **(Left)** Train accuracy curves. **(Middle)** Test accuracy curves. **(Right)** Train and test accuracy plotted against each other. The horizontal axis is train accuracy, and the vertical axis is test accuracy. Higher curves indicate better generalization for the same train accuracy

## A.4 FURTHER MIXER RESULTS AND ANALYSIS

Here, we attempt to understand further the Mixer (Ren et al., 2022) results shown originally in Table 1. This experimental setting is especially interesting since mixers have different optimization dynamics than regular MLPs. In fact, Ren et al. (2022)'s method achieves higher test accuracy than backprop despite being a gradient guessing method. Our results in Table 1 show a similar trend. To further understand this, we plot the train/test curves and compare the two methods under two conditions: with and without local losses. We see that the overall accuracy for each model is much lower without local losses, and the gap between $W^\top$ and activation perturbation is even larger since variance reduction matters more when the overall variance is larger. We also plot a scatterplot of train accuracy against test accuracy and find that for any given train accuracy, our models generalize better than backprop despite backprop achieving better training accuracy.

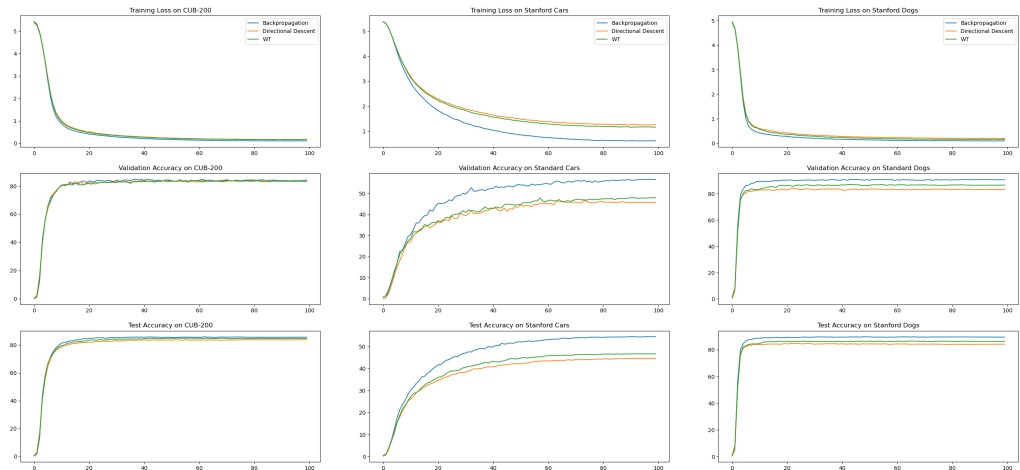

Figure 9: Gradient guessing strategies applied to Visual Prompt Tuning Jia et al. (2022) where prompt tokens are trained with backpropagation, directional descent, and $W^T$.

## A.5 APPLICATIONS IN FINE-TUNING LARGE MODELS

**Visual prompt-tuning experiments**: As models become ever larger, the burden of even instantiating the computational graph of a model for a single minibatch becomes untenable for modest computational resources. Backpropagation exacerbates this issue for tuning methods like Lester et al. (2021); Liu et al. (2022); Lu et al. (2021), which require intermediate states of the graph to be saved for the backward pass. We apply our gradient guessing methods to the space of parameter-efficient fine-tuning (PEFT), where memory-efficient training methods are relevant. One popular approach is prompt-tuning Lester et al. (2021); Jia et al. (2022). For pre-trained vision transformers (VIT-B/16 Dosovitskiy et al. (2020)), we tune the prompt tokens by guessing the gradient of the prompt. Figure Figure 9 shows that our $W^T$ method consistently outperforms directional descent for all fine-tuning datasets (CUB-200 Wah et al. (2011), Stanford Cars Krause et al. (2013), Stanford Dogs Khosla et al. (2011)).

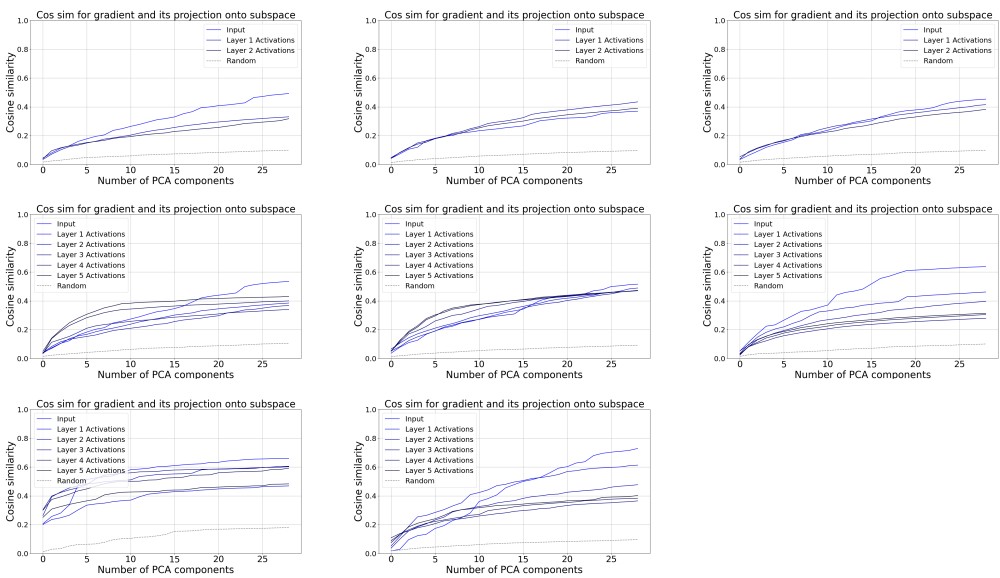

Figure 10: Activation and gradient subspace similarity

## A.6 ACTIVATION SUBSPACE PLOTS

**Activation subspace plots** The activation subpsace figure in the main paper superimposed curves from many different networks and epochs. Here we show some of them separately for various network widths, layers, and depths.

## A.7 MLP TRAINING PLOTS

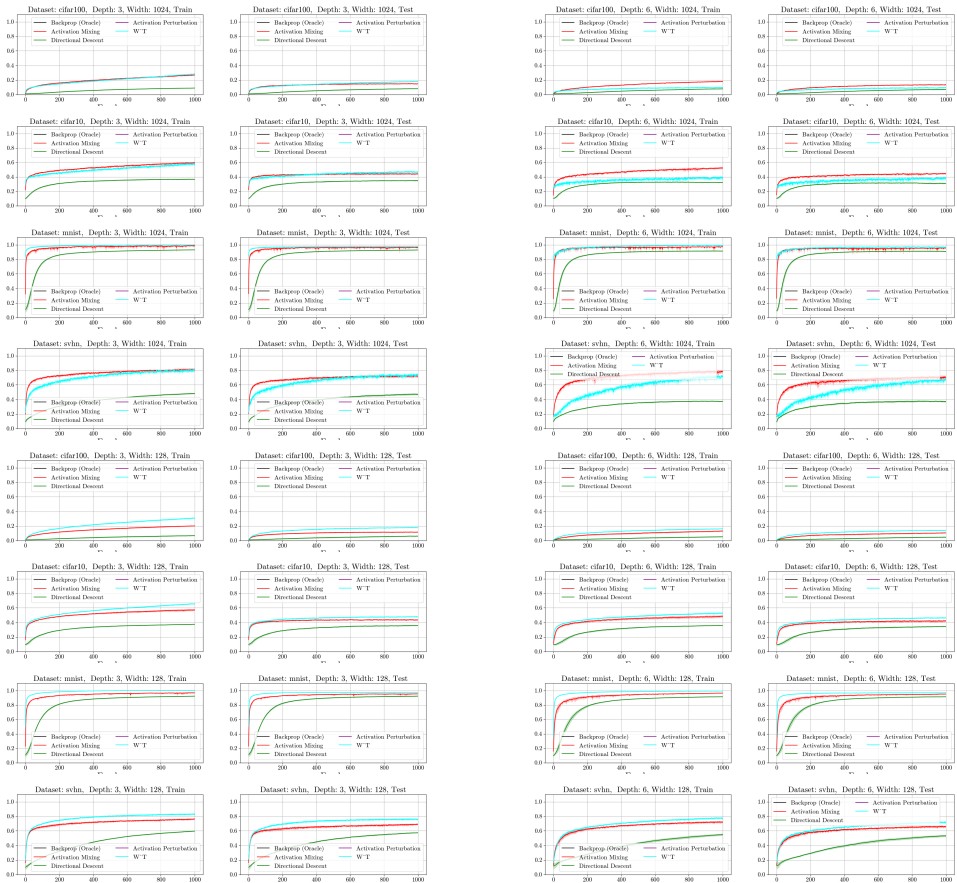

Figure 11: Train/Test plots for MLP training in Table 2.

