# OpenReview forum: "How to Guess a Gradient"
_ICLR.cc/2024/Conference — ICLR 2024 Conference Withdrawn Submission_

### Official Review · Reviewer_xG3Y · 2023-10-24

**Soundness:** 2 fair
**Presentation:** 2 fair
**Contribution:** 2 fair
**Rating:** 6
**Confidence:** 3

**Summary:**

The authors study the problem of optimizing a deep neural network without explicitly computing the gradient through backpropagation. This means that the gradient needs to be guessed. The main idea of the paper is that the gradients lie in a subspace that is lower than the size of the model, so the gradients should be guessed in this subspace. The authors show empirically that the gradient guesses obtained through this method are closer to the true gradient than previous methods, but they are biased estimators. They show that the performance of their methods improves with respect to previous methods, but it is still quite lower than using backprop. They argue that this is due to the bias in the gradient estimates and provide empirical evidence in support. Surprisingly, the authors also benchmark their method on the MLP mixer, and they obtain better performances than if they train with backprop.

I lean toward rejection because [I elaborate on these three points in the Weaknesses section]:

(P)resentation. The English is good and the paper starts well, but the presentation gets progressively poor and hard to follow. Especially toward the second part of the paper, there are disconnected paragraphs which it is hard to reconnect to the global story. Although the proposed concepts are not particularly hard, it takes more effort than necessary to understand how experiments are performed, and which evidence leads to what. Further, figures and tables could be much clearer than at the current state.

(S)oundness. Much of the told story is based on reasonable arguments, confirmed by empirical evidence. Since the empirical evidence is the basis of the paper's message, it should be better substantiated than it currently is. This goes along with the presentation, which makes it hard to understand how experiments were performed (e.g. it seems that no hyperparameter tuning was done, and that the evidence is based on single runs), and therefore to understand whether the methodologies are sound. The MLP-mixer runs contain a strong message, and yet nothing is provided to make this evidence strong.

(C)ontribution. A main selling point of the paper is not developed. This is, the good performance on the MLP mixer is just flashed and no attempt is done at analyzing it. This analysis is needed to make this finding consistent, and it would shed light on the argument presented by the authors, stating that it is the bias on the gradient that leads to low performances.

**Strengths:**

STRENGTHS
- The main idea of the paper, of reducing the dimensionality of the guessing space, is worth exploring. Although the results are not good nor fast nor general enough to replace backprop, it is still a step forward.

- The gradient guesses are much more aligned to the true gradient than previous methods. This is an improvement with respect to previous methods.

- Benchmarks on the MLP mixer show that in that case guessing the gradients is actually better than performing backprop. This is potentially very impactful, since it shows a (large-scale) case in which gradientless optimization matches backprop.

- The authors argue that one can solve the lower performance problem by eliminating the bias in the gradient guesses. Therefore, there is hope that the current limitations of the method can be overcome.

**Weaknesses:**

(HYPERPARAMETERS) Was hyperparameter (HP) tuning performed (and how? e.g. based on which metrics) for all the models or only for those of figure 7? And even then, the displayed learning curves are only on the training set: was the tuning performed on the training set? How many times was each run repeated? I see no error bars, which would lead me to think only once. Do the models have a regularizer term (I guess not, because it would influence the shape of the gradient)? Also, I think that the same HPs were used for different methods, but I believe that the comparison should be performed on the optimal choice of HPs, since likely the HPs that are good for one dynamics are not necessarily good for another.

(CODE) I could not find a statement saying that the authors provide their code, and would have liked to check it (at least to answer to some implementation questions which I was not able to find an answer to in the paper).

(COMPARISONS) The comparisons between methods are shown for a fixed number of steps, but the sheer number of steps is not comparable among methods. Learning curves (accuracy and loss as a function of training time) should be provided at least in the appendix. Could it be for example that the $W^T$ method would often perform better if we only let it evolve longer?

(MLP-MIXER)
The results on the MLP mixer are surprising to me. Why is the $W^T$ method performing worse with simple models, and all of a sudden it is doing better than backprop? Can the authors convince the reader that this is not a fluke? How many times were these runs repeated? Can the authors show the learning curves and describe in full detail the model training? Also, if the authors' argument on the bias is correct, then they should see that the gradient guesses on the MLP mixer have no bias, right? Why didn't the authors study this? This result can be very nice, but at the current state it feels incomplete.

(TWO-REGIMES) From figure 1, there seem to be two regimes in the dynamics. At the very beginning of learning, the cosine similarity is much higher than throughout the rest of the run. Do we have an intuition for this? Later on, when measuring the effect of the bias, the authors do it after one epoch. But this is in the initial regime where the cosine similarity is high, and not in the more interesting regime (which starts immediately after) where the cosine similarity becomes stable throughout the run.

(PRESENTATION-QUALITY)
Although the manuscript is written in a good English language, the presentation is not clear. The presentation starts clear, but it becomes progressively more fragmented, and it becomes very hard to follow the flow.

I list a series of additional specific points on the presentation:
- The figures and tables are scattered anywhere in the text. For example, table 2 is referenced (before table 1) in page 5 but appears in page 9; or figure 1, which is referenced after figure 2, appears on page 2 but is only referenced on page 5.
- Instead of "Please refer to the supplementary section" it would be nicer to know the exact section. Same thing when referencing a figure with multiple plots (e.g. write "fig.4-left", instead of just "fig.4")
- Y-labels are always defined, but sometimes in the caption, sometimes in the title, and sometimes on the y axis. I think that it would be nice to at least always have them on the y axis. Likewise, my understanding is that the left column of table 2 is training data, while the right column is testing data, that could be written on the tables (it's only vaguely written in the caption). If find it nice to put (ours) on the methods proposed by the authors, and would also put it elsewhere, e.g. in the table of Fig.1.
- Fig.1 Why are the curves of $W^T$ and activation perturbation missing on the left hand side plot?
- The black curves on many sets of figure 3 are not visible. I have to go all the way to the table on page 9 to find out that they are hidden below the bounding box.
- The self-sharpening is introduced as an effect but then it looks like it is a method, and then there is a discussion on SVD which it is not clear whether it was used for algorithmic or only descriptive purposes. I find the overall description confusing. In general, there I found no clear place where to find the definitions of the methods (except maybe for $W^T$, but it would be nice that all the methods be described consistently with one another)
- The math of section 2.2 is simple, but I find the way it starts very confusing. The formulas clarified the text for me - it should be the opposite. Also, when mentioning that hypotheses are made, please specifically list which hypotheses.
- The Visual prompt-tuning experiments in the appendix come out of the blue, and the short paragraph justifying it is not clear.
- labels are cut (fig.7) and legends require zooming a lot
- The term "guess" has a technical meaning in this paper, so I would use a couple of words to define it.
- Symbols. E.g. the usage of the $\top$ symbol is not consistent between Eq.5 and the following paragraph (and in general throughout the text). Or when the weights W1,...,Wk, it is left to the reader to understand these are matrices corresponding to each layer, and not all the weights of the model (so k is the number of layers, and not the number of weights). Or the expectation in equation 14 is over the guesses of the gradient, which is why it does not act on $\nabla L$.
- The acronym JVP is defined in the appendix but used in the main paper
- It would be much easier if the numbered list at the beginning of section 3 corresponded with a number at the beginning of each sectin where those points are treated.
- typo: there is a period missing at the end of section 3
- typo: "This bias is further in the next experiments and Section 4."
- typo: linear combinations OF all the training example activations in the batch

(BREADTH) Although I acknowledge that it is interesting to find alternatives to backprop even though they are not at the level of the state of the art, the current findings only apply to MLP models with ReLU activations, and are only based on heuristical arguments. Additionally, these methods are much (but how much?) slower than using backprop, and significantly worse. The current method relies on specific knowledge of the architectural details, so even if it was faster and better performing than backprop, it would still probably not be implemented.

**Questions:**

- For the related literature section: that learning happens in a tiny subspace was already known since years (https://arxiv.org/abs/1812.04754). Interestingly, this subspace is even smaller than the one used by the authors, so this can perhaps be used for further improvements of the proposed method.
- The discussion section discusses open research lines. I would add some references.

- I do not understand why the first term of Eq.11 can be ignored. Unless the weights are initialized in the origin (which is usually not the case), this term should be big

- Why is self sharpening only used for the W:1024 models?

- I would like to see a table of how long each of these methods (including backprop) take, per gradient guess, and also how many more epochs are needed to train (we need the training curves here).

---

> ### Author Response · Authors · 2023-11-23
>
> Thank you for taking the time to review our submission and providing thoughtful feedback. We appreciate you highlighting the strengths of our work. We also agree with your assessment that this work is not intended to be immediately practical, but rather a step forward.
>
> **Error bars**: We have added the error bars for Table 2. Specifically, we report mean and standard deviation over 5 runs. We will soon add error bars for Table 1 as well.
>
> **Training/Testing curves**: We have added a figure in the supplementary section depicting the train accuracy and test accuracy curves for each model in Table 2. Each plotted curve is the mean of 5 runs, with translucent bands around it indicating standard deviation.
>
> **Code**: We will share our code and model checkpoints along with the camera-ready draft for transparency and reproducibility. If you have any specific questions about the code, we can answer those as well.
>
> **MLP-Mixer**: We have added train/test curves and more experiments isolating the reason for the increase in accuracy (see figure 8 and explanation below). The result is repeatable, and we will add error bars in Table 1 soon.
>
> **Why is the method better than backprop on the Mixer architecture?**: Please note that Ren et al’s [1] Mixer model (using their local losses and activation perturbation) also achieves higher test accuracy than backprop. We only replace the activation perturbation with W^T, which results in a further 1.5% boost in test accuracy. The core factors behind this sudden improvement are twofold: (1) The local losses used by Ren et al provide much more supervision and help the model train to a high accuracy; (2) Their Mixer architecture seems to overfit with backprop.
>
> To isolate local losses as the factor, we ran the same experiment without local losses and plotted the results in Figure 8. We see that the overall accuracy for each model is much lower without local losses, and the gap between W^T and activation perturbation is even larger (since variance reduction matters more when the overall variance is larger).
>
> We also plot train vs. test accuracy, showing that backprop gets better train accuracy (as expected) but generalizes worse. It is not in the scope of our work to speculate why Mixer architecture overfits with backprop, but this could be interesting follow-up work since gradient approximations traditionally overfit more [2].
>
> **“Are gradient guesses on MLP mixer unbiased?”**: The efficacy of MLP mixers in achieving high accuracy doesn’t necessarily imply unbiased gradients. This complexity is highlighted in our self-sharpening experiments, where significant bias accompanied high cosine similarity and training accuracy. Additionally, the unique dynamics of MLP mixers add another layer of complexity. For instance, the absence of gradient flow between blocks in MLP mixers reduces the potential for positive feedback loops. While bias clearly impacts MLP experiments, its interaction with the model dynamics is still not well understood. Our ongoing research aims to understand these complexities. Along these lines, we have also added further analyses to the supplementary section (Figure 7).
>
> **“MLP Mixer experiments/analysis feel incomplete”**: We appreciate your feedback, and would like to improve the experiments/analysis. Based on our reply so far, which additional experiment/analysis would you recommend?
>
> **“Two Regimes”**: In our understanding, the cosine similarity changes as the model trains, and the largest effects of that are seen right at initialization (and immediately after). As mentioned in the paper, our reported cosine similarity and 1-step effectiveness (now with error bars) are computed after each epoch and averaged over 50 epochs. Thus, we largely average over the second regime. This can also be seen by visually comparing the cosine similarity in the plot against the reported mean.
>
> **Writing quality and references**: We really appreciate the suggestions. We have incorporated several of these already and will follow all of them for the camera-ready draft. Our goal is to make the paper approachable and easy to read.
>
> **Proposed methods are heuristic; limited to MLP+ReLU**:  The observation that our methods are primarily demonstrated using MLP+ReLU models is valid and appreciated. However, it's important to note that our research, while using MLPs with ReLU activations as a foundational model, actually delves into a broader principle. The concept of partial backpropagation, utilizing locally available Jacobian information, is not confined to this specific architecture. In fact, this approach has the potential to be adapted to various architectures. The insights and findings presented in our paper, therefore, serve as specific instances of a more universally applicable principle. We believe this flexibility and adaptability are significant strengths of our proposed methods.

---

> > ### Author Response · Authors · 2023-11-23
> >
> > **Proposed methods are slower/worse**: While it is true that our methods don’t converge as fast or generalize as well (except Mixer experiments), the more interesting point is that *they work at all*. We are not proposing a replacement for backprop. If the goal is to create a replacement, it would likely contain a combination of local losses and our guessing strategy rather than our methods on their own.
> >
> > **Hyperparameters**: For our MLP experiments, we matched batch size, learning rate, number of epochs, and weight decay between all models. Here is our reasoning behind each such choice:
> >
> > Weight decay: always set to 0 since we did not want to interfere with the gradient statistics.
> > Batch size: always set to 512 since it affects gradient variance, and different batch sizes for different methods would give an unfair advantage to some methods.
> > Learning rate: always set to 0.0001 so we can do an apples-to-apples comparison of convergence. As noted in [3], the loss reduction per step is a function of the step size and gradient variance, so selecting different step sizes can obfuscate the effect of variance.
> > Epochs: always set to 1000. In our experiments, we found that most methods achieved their highest test accuracy long before 1000 epochs, so we considered that to be a sufficiently long time period. In principle, we can train some models for 10x longer, and then W^T would achieve 90%+ training accuracy, but its test accuracy would not improve.
> >
> > If this reasoning is incorrect, please let us know which hyperparameters you would like us to tune.
> >
> > **Figure 6, learning rate sweep**: This figure's goal was to show convergence rate differences under ideal learning rates, which is why we show training curves and pick the best learning rate for each method. As mentioned in A.1 (Learning rate and optimizer sweep), the metric is train accuracy at the end of 1000 epochs. The learning rates were optimized over the mean of 5 runs, but we show only one run. We will also add the error bars for these in the next few days.
> >
> > **Tiny subspace of gradient descent**: Thank you for the reference. This line of work, including the intrinsic dimensionality work we cited, was our inspiration for activation mixing. We hope to explore this further in follow-up work (e.g. by using Hessian eigenvectors directly as guesses).
> >
> > **Ignoring the first term in Eq. 11**: Thank you for pointing that out. We agree that this term is not necessarily close to 0, especially if the model has not been initialized at the origin. We have added that clarification to the draft.
> >
> >  Analytically, this term can be safely ignored when weight decay is used since it gets multiplied with an exponentially decreasing factor (we will include the formal proof in the camera-ready draft). We should note that our MLP experiments do not use weight decay, so this explanation does not apply. Instead, our justification is experimental. As shown in Figure 2 (right), the activation and gradient subspaces are well-aligned even though we ignore this term.
> >
> > **How long each method takes**: We have added training curves to the supplementary section (Figure 11). As for the time taken for each guess, this is hard to calculate precisely since our implementations are not optimized. We track each activation, each gradient, each weight’s rank, each weight’s SVD, etc. That said, each method is roughly 3x as much computation compared to backprop.
> >
> > [1]: https://arxiv.org/abs/2210.03310
> > [2]: https://arxiv.org/abs/2206.00823
> > [3]: https://arxiv.org/abs/1812.06162
> >
> >
> > Please let us know if there are any remaining questions or comments. Your feedback has helped us improve the paper significantly

---

> > > ### Comment · Reviewer_xG3Y · 2023-12-03
> > >
> > > I thank the authors for considering the points I raised.
> > >
> > > Several of my concerns on the presentation were addressed, and the authors acknowledge that they will address the remaining ones for the camera-ready version. Given the short times involved, I find it ok. Some additional information on the MLP-mixer and a related rationalization is provided. The improvements in the presentation make the results more clear what was exactly done, and the authors will provide their code. I thus increased my score accordingly.
> > >
> > > Some additional minor points.
> > >
> > > - The authors specify that the curves are averages over 5 runs, but I could not find this in the text. I would suggest to explicitly state this.
> > > - There is still some notation confusion between $^T$ and $^\top$. E.g. "(3) $W^\top$, which multiplies an isotropic guess by", which uses a different notation that fig.1-right.
> > > - The xlabels in fig6 are hidden.
> > > - Fig6: please write "training accuracy on the ylabels", or at the very least in the caption

---

### Official Review · Reviewer_Nqba · 2023-11-01

**Soundness:** 3 good
**Presentation:** 3 good
**Contribution:** 3 good
**Rating:** 6
**Confidence:** 4

**Summary:**

In this paper, in order to guess a better gradient (with less variances), the authors unfold the backpropagation and show that the gradient can be guessed in a much lower dimensional subspace and achieves better cosine similarity compared to the directional gradient. With the better-estimated gradient, they show that they can train a model with higher accuracy without backpropagation than previous methods and extend to cifar10 results.

**Strengths:**

The paper is easy to follow and the proposed method is intuitive. In addition, the empirical results are good compared to previous baselines. The idea is simple but effective.

**Weaknesses:**

How large is the largest model? Since the paper mentioned that the proposed method makes the training model with millions of parameters feasible, can we have some of these results?

Can we apply the methods to convolution networks or transformers? Is that possible or easy?

**Questions:**

See weakness

---

> ### Author Response · Authors · 2023-11-23
>
> Thank you for the review! Here are the answers to your questions:
>
> **Our largest models for MLP experiments**: Our largest MLP has 6 layers of width 1024, and operates on 32x32x3 images (for CIFAR10). Thus, it has 7.3 million parameters. Specifically:
>
>
> 3x128 MLP: 411k
> 3x1024 MLP: 4.2M
> 6x128 MLP: 460k
> 6x1024: 7.3 M
>
>
> **Our model for Mixer architecture (table 1)**: 918k parameters
>
>
> Crucially, we note that our method can train much larger models than directional descent, and in some cases, achieves comparable test accuracy to backprop (i.e. 77% test accuracy on CIFAR10 with Mixer architecture). This is a direct result of our gradient estimates’ cosine similarity being hundreds of times higher than a random guess.
>
>
> **Applying to CNNs/Transformers**: Our method can be applied to CNNs and transformers, but the current formulation does not utilize their architecture/structure. As a result, even though our method performs well on the Mixer architecture, it currently underperforms on CNNs. We hope to close this gap in follow-up work.
>
> Please let us know if you have any other questions or comments

---

### Official Review · Reviewer_58v1 · 2023-11-02

**Soundness:** 3 good
**Presentation:** 3 good
**Contribution:** 3 good
**Rating:** 6
**Confidence:** 4

**Summary:**

The authors propose several improvements over random directional derivative approaches to approximate the gradient without back-propagation.

These are based on considering the gradient calculation and leveraging properties of the components of this gradient calculation - to reduce the variance / make better random samples for parts of approximating the gradient.  E.g., instead of randomly sampling a direction, randomly sampling part of the gradient calculation with a specific structure and plugging in the rest of the formula to estimate the gradient, and generating the random samples in a particular way, e.g., from a subspace more likely to contain the true gradient / closer to the subspace of the true gradient.

They compare the proposed methods to baselines and a prior enhanced method, and show significant improvements across many architectures and several datasets, analyze and discuss the results and take-aways in detail.

**Strengths:**

-The ideas proposed to improve gradient approximation seem novel and are interesting - I feel they could be useful and motivate other work

-The ideas are well motivated and introduced, and overall the paper is well-organized and flows nicely

-Extensive experiment results are provided

-Good discussion and analyses is provided including limitations

**Weaknesses:**

I feel many of these weaknesses listed below could be addressed, and I would consider raising my score in that case.


1) No discussion of broader related work.  I.e., the authors only mention specifically work on using the directional derivatives, but there is a much broader area of work on non-gradient learning for neural networks.  It would be best to point to this other work and situate this work in context of the broader work in this area (e.g., in a small related work section for example).

For example, one work in particular that comes to mind and is similar in spirit to use random directions, is using random perturbations.  Simultaneous perturbation stochastic approximation (SPSA) has been used, but similarly to improvements in this work and others for the directional derivative approach, a likelihood ratio approach has been proposed, which generates random samples from a more appropriate distribution.  I would be very interested to see a discussion of how the current work and focus area compares to this other work (especially since this work has shown potentially better results than the directional derivative approach).
Example papers:
- "A New Likelihood Ratio Method for Training Artificial Neural Networks" https://papers.ssrn.com/sol3/papers.cfm?abstract_id=3318847
- "One Forward is Enough for Neural Network Training via Likelihood Ratio Method" https://arxiv.org/abs/2305.08960 (posted earlier in the year as "Training Neural Networks without Backpropagation: A Deeper Dive into the Likelihood Ratio Method")

There are also other interesting approaches, such as:
- "The Forward-Forward Algorithm: Some Preliminary Investigations" https://arxiv.org/abs/2212.13345
- "Evolution Strategies as a Scalable Alternative to Reinforcement Learning" (more generally evolutionary algorithms) https://arxiv.org/abs/1703.03864




2) There are some technical issues, missing details, and incorrect or unsubstantiated statements

- this line in the intro makes no sense:
"...and proposes addressing it by augmenting the network supervised and unsupervised loss functions located near trainable parameters."
what are you trying to say?

- This is incorrect: "Our second insight is that by the very nature of ReLU activations, ∂ReLU(si)/∂si will typically be a sparse diagonal matrix, which will “zero out”"
s_i is not a matrix, but a vector - so there is no way for the gradient with respect to s_i to be a diagonal matrix.  Furthermore, for what reason should we expect this to be sparse?  It's never stated.  In my experience, this depends on a lot of factors and is not necessarily the case.

- "our third insight is that W ⊤ often effectively has low rank. " - this is unsubstantiated - cite a reference or explain / prove why this is the case.

- What is the "activation perturbation baseline" - it's shown in results and mentioned as a baseline but it's never explained what it is.

- The "self-sharpening" method is not completely / clearly described / explained

- Not all method results are reported in all experiments.  It seems only a subset of method results are shown in several cases, without explanation.
-- In particular, the prior work being compared to should be included in more experiments, e.g., in the detailed results of Table 2.


3) writing could be cleaned up and improved...

**Questions:**

Please also see the questions listed in "weaknesses" section.

Typo: "forward-more differentiation" in Intro

Why not combine the 3 methods?  It's not completely clear that they are separate at first reading

---

> ### Author Response · Authors · 2023-11-23
>
> Thank you for the review!
>
>
> **Related work**: We completely agree, and will include a formal related work section in the camera-ready draft. We hope to create an approachable and complete summary of the developments in this subfield.
>
>
> **Likelihood ratio approach**: The likelihood ratio approach is especially interesting as it seems to be compatible with many different distributions. Since our guesses follow the multivariate normal distribution, this seems like a good match. Furthermore, the probabilistic formulation also naturally supports techniques like importance sampling which could help us reduce the bias significantly.
>
>
> **Confusing line in intro**: Thanks for pointing out the confusion. That line refers to local losses used by Ren et al. We have re-written that line to make it more clear.
>
>
> **Relu derivative**: ∂ReLU(si)/∂si refers to the jacobian of relu activations with respect to the inputs to relu (pre-activations). Since the input and the outputs to ReLU are both vectors, the jacobian is a square matrix, and since each input controls one and only one output, it is a diagonal matrix. We have edited that section of the draft to explain this a bit better.
>
> **Sparsity**: Thank you for the correction. It is true that the ReLU activations may not be too sparse, and it depends on many factors. For example, at initialization, close to 1/2 of the units are active. While this is still useful for reducing the guess space, we have clarified this statement in the revised draft. For reference, gere is a plot of observed sparsity fractions in practice for a 1024x6 MLP on CIFAR-10:
>
>
> **Sparsity fraction**: https://imgur.com/a/IsGUE3j
>
>
> **W^T low rank**: Thank you for pointing this out. Here is previous work showing the claim [1], and we have also added this to the draft.
>
>
> **Activation Perturbation baseline**: This baseline refers to an isotropic random guess in activation space rather than weight space. This method is also used by Ren et al. We have added this clarification in the revised draft.
>
>
> **Self-Sharpening**: This is an unexpected phenomenon we observe during some of our guessing methods where the space of guesses becomes more narrow or ‘sharpens’, and this improves the cosine similarity to the exact gradient. The reason we call it ‘self’ sharpening is because this narrowing is caused by a feedback loop of decreasing rank of downstream weights. This decreasing rank narrows the guess space, which makes updates less diverse and further lowers the rank. We have updated the explanation in the draft, and will further improve it for the camera-ready draft.
>
>
> **Reporting all methods**: Could you please clarify what experiment combinations we should include for Table 2? We can add them to our revised draft for completeness. Related to Table 2, we have also added the corresponding training/testing curves. This covers our proposed methods and relevant baselines (directional descent, backprop, activation perturbation) for various MLP sizes.
>
>
> **Writing quality**: Thank you for the feedback. We will focus on writing quality and clarity for the camera-ready draft.
>
> [1]: https://arxiv.org/abs/2103.10427

---

### Official Review · Reviewer_Gf7u · 2023-11-04

**Soundness:** 3 good
**Presentation:** 4 excellent
**Contribution:** 3 good
**Rating:** 5
**Confidence:** 4

**Summary:**

This paper investigates the problem of guessing gradients for forward automatic differentiation or "forward gradients" methods.
The main focus is to reduce the variance of the forward gradient estimator by generating guesses that possess provably better alignment with the true gradient.
Most of the presented improvements are agnostic to data and depend only on the network structure, except for the activation mixing method, which indeed relies on the forward propagation of data samples in the model.
The paper is well-written, pedagogical, and the experiments are accompanied by theoretical insights.
Overall, this is a very interesting investigation that should be valuable to research on optimization with forward gradient methods.
The experimental protocol is relevant, except for the choice of the model being evaluated, since its accuracy on standard classification benchmarks is extremely low.

It is worth noting that the authors acknowledge in the main body that their proposed improvements do not make forward gradient methods an alternative for standard backpropagation, and that further research is needed for such approaches to become competitive.

**Strengths:**

1) The experimental protocol is relevant, the theoretical insights are valuable, and the different experiments bring valuable knowledge about the practical training dynamics.
2) The bias of the proposed methods is appropriately characterized, and its relation with respect to the drop in performance is appropriately showcased. In particular, the authors show that it is a crucial limiting factor for good performances, even in cases where the cosine similarity between the guess and the true gradient is high.

**Weaknesses:**

1) The test accuracies obtained with backpropagation in Table 2 are extremely low. The model clearly overfits the dataset, which is not surprising for an MLP.
2) It is not clear from the text whether the self-sharpening phenomenon is a desirable property of the training dynamics. As far as I understand, it is not.

[Minor comment]
3) At the end of the paragraph "Effect of bias on our methods", the sentence "This bias is further in the next experiments and Section 4." has a missing word.
4) The second paragraph of "SUMMARY AND DISCUSSION" begins with "another interesting future research..." while this is the first research direction proposed. I would switch the order between the second and third paragraphs since the bias of this method has been identified as the main limiting factor.

**Questions:**

1) Could the authors replicate some of their experiments with another model that has an accuracy much closer to SOTA than chance level on these datasets?
This is particularly important for Table 2. The best accuracy for CIFAR10 is 53.5%. Similarly, it is 83.5% for SVHN. This is unacceptable for such a research paper. Not that the reviewer has some particular taste for some particular model, but it is impossible to adequately judge the practical relevance of these contributions if the accuracy is exactly half-way between SOTA and chance level. The SOTA for CIFAR10 is > 95% while chance level is 10% - your model is 53.5% which is very problematic. I am not saying that the proposed method should all yield > 90%, but it should be the case for the baseline (standard backpropagation).

2) How is the 1-step effectiveness calculated? Has it been calculated for 1 mini-batch or is it the average for multiple training steps?
It would be nice to have a mathematical formula to understand what is at stake here.

3) If the self-sharpening in the $W^{\top}$ setting is due to the matrix $W$ becoming low rank, is the self-sharpening phenomenon a desirable thing because it leads to higher cosine similarity, or should someone avoid this dimensional collapse because it hampers generalization?

4) An interesting alternative gradient guessing scheme is presented in [1] where a gradient from a small auxiliary classification network is used as a guess direction. It would be nice to be able to compare a purely structural bias such as proposed in the presented work, where the bias is strongly data-dependent. (The experimental settings are quite different, making a fair comparison difficult.)

[1] - L. Fournier, S. Rivaud, E. Belilovsky, M. Eickenberg, E. Oyallon. ICML 2023. Can forward gradient match backpropagation?

---

> ### Author Response · Authors · 2023-11-23
>
> Thank you for the review. We deeply appreciate your interest and suggestions. We have fixed the writing as you suggested, and will address your questions and mentioned weaknesses below:
>
>
> **1-step effectiveness**: This metric uses only 1 step rather than multiple training steps, but we average that metric over the entire dataset. Mathematically, we are computing:
>
>
> Drop_method = loss_original - loss(w - eta * estimated grad)
> Drop_backprop = loss_original - loss(w - eta * true grad)
> Effectiveness(method, eta) = drop_method/max(drop_method, drop_backprop)
> Highest_effectiveness(method) = max_{eta \in [1e-6, 1e-5, … 1e-1]} Effectiveness(method, eta)
>
>
> That is, we are computing how much the loss goes down with the estimated gradient compared to the true gradient. Unlike cosine similarity, this metric relies on the local curvature of the loss landscape too. This makes it more useful than simple cosine similarity since some errors/biases may not be relevant to the loss (e.g. flat directions in the loss landscape). We average this metric over the entire dataset and report its mean and standard deviation.
>
>
> **Self-sharpening phenomenon**: We find self-sharpening to be an unexpected and potentially useful phenomenon (in the context of narrowing guess space) that emerges from the guess bias. We opted to simply report this phenomenon and did not try to control/regulate it in our experiments, and it causes the model to overfit when completely unregulated. However, our singular value experiments show that the rate of self-sharpening can be finely controlled (Figure 5), and we hope to exploit controlled self-sharpening in follow-up work to get fast convergence and less collapse/overfitting.
>
>
> **Fournier et al**: Thank you for sharing this reference. We will add this to the related work section in the camera-ready version. Their thorough experimental analysis is also a useful template for our evaluations, and combining the two methods might produce interesting results. Although we cannot compare against this work in the time frame of this rebuttal, we hope to do a larger scale comparison on multiple architectures in follow-up work.
>
>
> **90% test accuracy with backprop**: We agree that a higher accuracy model is important to understand this method’s effectiveness fully. For reference, our highest accuracy on CIFAR10 (77.4% test, 90% train) is on the Mixer architecture/losses used by Ren et al. This architecture/loss fails to achieve 90%+ test accuracy with backprop, at least for the model sizes described in the paper (0.9M, 76.4% test accuracy).
>
>
> We tried to get higher test accuracy with backprop by varying the numbers of blocks and groups, augmentations, and training schedules for the Mixer architecture. However, we were unable to achieve 90%+ test accuracy. To our knowledge, popular MLP-Mixer implementations rely on pre-training and/or significant data augmentation like CutMix to excel at vision datasets like CIFAR10 [1,2].
>
>
> Thus, we also evaluated our method on VGG16 without batch-norm or dropout. We also added random image cropping (padding=4), horizontal flipping, and mixup augmentation. Although this architecture is not an ideal fit for our method, it gets 90% test accuracy with backprop. The preliminary results are:
>
> |       | Train | Test |
> | ----------- | ----------- | ----------- |
> | Backprop     | 97.9%      | 90.0% |
> | W^T   | 28.7%       |  31.2% |
> |Directional Descent | 14.2% | 14.7% |
>
> W^T currently underperforms in this setting because the current implementation does not exploit any CNN structure or benefit from local losses like in Ren et al. We also have not done any hyperparameter tuning. We are working on those and will keep you updated.
>
> [1]: https://github.com/omihub777/MLP-Mixer-CIFAR
> [2]: https://github.com/sayakpaul/MLP-Mixer-CIFAR10

---

> ### Comment · Reviewer_Gf7u · 2023-11-23
> **Further questions**
>
> I thank the authors for their reply.
> Here is my response:
>
> **1-step effectiveness**: I thank the authors for providing a formal definition of the formula used to compute their effectiveness score, and particularly for emphasizing its superior relevance over cosine similarity.
>
> **Self-sharpening phenomenon**: I completely understand that the authors discovered an interesting phenomenon in the training dynamics of forward gradient methods, and reported it for further investigations. I am not asking if the authors *claim* that it is a desirable phenomenon. I am rather asking if, at the time the authors are submitting the article, given the combined experimental results of the activation mixing experiments where the self-sharpening phenomenon revealed itself, and the experiments where the singular value distribution of the matrix $W^\top$ are artificially manipulated, does this phenomenon seem desirable or not?
> It is not clear from the manuscript.
> My understanding is that it is not, since it seems to drive the model towards an optimization path where the effective rank of the operator $W^\top$ collapses and thus loses statistical capacity.
> Again, I am not asking the author to provide a definitive answer to this research question, but rather to summarize the conclusion they can draw from the experiments conducted so far, and if possible, guidelines on what should be the next research step.
>
> **Fournier et al**: It would be particularly interesting to combine both methods and ablate the contribution of each individual trick, especially since the reported figures are quite high and the inductive bias (greedy optimization) of their method is the main limiting factor, preventing their gradient guess space to cover the true gradient space, hence resulting in poor cosine similarities.
>
> **90% test accuracy with backprop**: These results are very interesting even though not in favor of the proposed method yet.
> I do not have much experience with MLP architectures on small-scale datasets, but I know they are avoided in practical applications because of their overfitting tendency. As a first workaround, the authors could use an MLP-mixer pre-trained on ImageNet, albeit there might be some dimensionality issues to handle.
>
> It is however surprising that VGG16 performs so poorly with your method and simple directional descent. I would actually expect something closer to 40-50% with directional descent alone, whether activation or weight perturbation is used. I think the author should have run the VGG-16bn version which contains batch-norm as it is known to greatly stabilize training, allowing a wider range of learning rates without numerical issues. Likewise, I would also suggest making experiments including dropout since you could only perturb the units that are not set to zero, in a spirit similar to how ReLU activations are being handled. Preventing overfitting does solve the fact that the training accuracy remains very low. But the fact that batch-normalization stabilizes training might be explained by a smoother loss landscape, which would make optimization less challenging, thus yielding performances favorable to your method. It would also be interesting to know if the self-sharpening phenomenon remains in such a setting, thus emphasizing a consistent artifact of the proposed training dynamics.

---

### Author Response · Authors · 2023-11-23
**Thank you for the feedback**

We sincerely appreciate the reviewers' feedback, and have incorporated much of it into the current draft. We hope to keep improving it further for the camera-ready version. In the meantime, here is a summary of the major changes:

- Added further analyses for the bias, its interaction with momentum, and how multiple layers affect the guessing space
- Added further analyses to help understand why Mixer models achieve high accuracy.
- Edited main text for clarity and readability
- Edited figures for readability, added all the baselines to Figure 1
- Added error bars and plotted train/test curves for MLP experiments.

We look forward to hearing any additional suggestions/comments.